# Characterization of Tapeworm Metabolites and Their Reported Biological Activities

**DOI:** 10.3390/molecules24081480

**Published:** 2019-04-15

**Authors:** Phurpa Wangchuk, Constantin Constantinoiu, Ramon M. Eichenberger, Matt Field, Alex Loukas

**Affiliations:** 1Centre for Molecular Therapeutics, Australian Institute of Tropical Health and Medicine, James Cook University, Cairns, QLD 4878, Australia; matt.field@jcu.edu.au (M.F.); alex.loukas@jcu.edu.au (A.L.); 2College of Public Health, Medical & Vet Sciences, James Cook University, Townsville, QLD 4811, Australia; constantin.constantinoiu@jcu.edu.au; 3Institute of Parasitology, Vetsuisse faculty, University of Zurich, Winterthurerstrasse 266a, CH-8057 Zurich, Switzerland; ramon.eichenberger@uzh.ch; 4John Curtin School of Medical Research, Austalian National University, Canberra, ACT 2600, Australia

**Keywords:** Helminths, tapeworm, excretory-secretory products, small molecules, bioactivities

## Abstract

Parasitic helminths infect billions of people, livestock, and companion animals worldwide. Recently, they have been explored as a novel therapeutic modality to treat autoimmune diseases due to their potent immunoregulatory properties. While feeding in the gut/organs/tissues, the parasitic helminths actively release excretory-secretory products (ESP) to modify their environment and promote their survival. The ESP proteins of helminths have been widely studied. However, there are only limited studies characterizing the non-protein small molecule (SM) components of helminth ESP. In this study, using GC-MS and LC-MS, we have investigated the SM ESP of tapeworm *Dipylidium caninum* (isolated from dogs) which accidentally infects humans via ingestion of infected cat and dog fleas that harbor the larval stage of the parasite. From this *D. caninum* ESP, we have identified a total of 49 SM (35 polar metabolites and 14 fatty acids) belonging to 12 different chemotaxonomic groups including amino acids, amino sugars, amino acid lactams, organic acids, sugars, sugar alcohols, sugar phosphates, glycerophosphates, phosphate esters, disaccharides, fatty acids, and fatty acid derivatives. Succinic acid was the major small molecule present in the *D. caninum* ESP. Based on the literature and databases searches, we found that of 49 metabolites identified, only 12 possessed known bioactivities.

## 1. Introduction

Tapeworms are platyhelminth parasites that cause devastating diseases in both humans and livestock, such as neurocysticercosis and hydatid disease [1]. Human infections with tapeworms occur when people eat raw or undercooked infected beef and pork or by ingesting tapeworm eggs. Humans also occasionally get infected with the dog/cat flea tapeworm, *Dipylidium caninum*, by ingesting the cysticercoid stage which is found in dog or cat fleas (*Ctenocephalides* spp.) [2]. The method of transmission to humans is usually a result of close contact between children and their infected pets. In the small intestine of the vertebrate host the cysticercoid develops into the adult tapeworm which reaches maturity about one month after infection. The adult tapeworms measure up to 60 cm in length and resides in the small intestine where they attach to the gut wall using their scolex. The adult tapeworm releases proglottids (or segments) from its terminal end as they become gravid. The proglottids then migrate to the anus and are passed in the stool. Most *D. caninum* infections of dogs, cats and humans are asymptomatic and the infection is readily treated with praziquantel [1,2].

While these parasites feed in the small bowel, the adult tapeworms actively release excretory-secretory products (ESP) into the mucosa to modify their environment and circumvent the immune response of the host [3]. Tapeworm ESP have been shown to suppress inflammation by inducing regulatory B cells [4], tolerogenic dendritic cells [5], and regulatory T cells [6] which produce suppressor cytokines that keep inflammatory T cells and their effector molecules under control. These immunoregulatory strategies protect gastrointestinal tapeworms (such as *D. caninum*) from being ejected by the immune response but also provide protection to the host (only when present in moderate numbers) against gastrointestinal inflammatory diseases that result from a dysfunctional immune system. For example, the high molecular weight extracts of the rat tapeworm, *Hymenolepis diminuta*, has been shown to protect mice against experimental colitis and arthritis [7,8].

Helminth ESP comprise mainly proteins, nucleic acids, glycans, lipids, extracellular vesicles and small molecules (SMs), which are released in a variety of “packages” to achieve a range of biological functions. To date, most attention has been afforded to characterising the protein components of ESP from helminths, and we know remarkably little about their non-protein SM components [9,10]. SM are responsible for many essential biological functions. Where investigated, lipids and glycans produced by helminths are restricted to different developmental phases of the parasite’s lifecycle, implying tailored expression profiles for distinct functions in different environmental niches. For example, in the human gastrointestinal nematode *Ascaris lumbricoides*, ecdysones (steroidal prohormone) facilitate molting, and dimethylheptatriacontanes are involved in sexual communication [11]. Understanding the overall profile of helminth SM metabolites, which can be characterized using metabolomics techniques, will facilitate development of new tools for the diagnosis and treatment of helminth infections, as well as the development of novel therapeutics for treating inflammatory diseases [12]. Understanding metabolite expression patterns and identifying the SM components of ESP could provide important information on the molecular basis of parasitism and shed light on the various physiological processes that govern host-parasite interactions [9]. 

We recently characterised the secreted and somatic SM complement of gastrointestinal hookworms and showed that these worms secrete SM that suppress inflammation in a mouse model of inflammatory bowel diseases and reduce inflammatory cytokine secretion by human peripheral blood mononuclear cells [13]. In this study herein, we have used gas chromatography mass spectrometry (GC-MS) and liquid chromatography mass spectrometry (LC-MS) to characterize the ESP metabolome of *D. caninum* and searched the literature and public databases for reported biological functions/roles/properties of the identified SM, with a particular focus on anti-inflammatory properties. 

## 2. Results

### 2.1. Polar Metabolites of D. caninum ESP Identified Using GC-MS 

Adult *D. caninum* were collected from dogs and were cultured in a single component glutamax culture medium with antimycotic/antibiotic. Glutamax is a simple medium that consists of 200 mM l-alanyl-l-glutamine dipeptide in saline. Culture medium containing ESP was filtered (<10 kDa centrifugal filter) to obtain small molecule extracts, lyophilized, fractionated and then analysed using GC-MS and LC-MS. While GC-MS was used for identifying targeted polar and fatty acid components, LC-MS was used for analysing the targeted short chain fatty acids (SCFA). The ESP of *D. caninum* was fractionated to obtain a biphasic partition of the solution: upper aqueous phase (methanol:water) and lower organic phase (chloroform), which were derivatized and analyzed for polar and non-polar metabolites using GC-MS and LC-MS. 

For GC-MS analysis, an Agilent’s Mass Hunter Quantitative Analysis software (v.7), Mass Hunter Library (MHL), in-house Metabolomics Australia metabolite library (MAML), NIST library and the Metabolomics Standard Initiative (MSI) level 1 was used for identification and validation of compounds. Each EI-MS spectra [M^+^] of a metabolite was matched with EI-MS spectra in the compound library. The MAML was developed by running authentic standards in the GC-MS prior to running test samples. This same method was used for running our samples in the GC-MS. ^13^C-sorbitol and ^13^C^15^N-valine were used as internal standards, which helped in determining the relative concentrations of each metabolite by comparing the metabolite peak areas with the peak areas of the internal standards. Using these techniques, we identified 35 metabolites from the polar fraction (Table 1) of *D. caninum* ESP. The top 10 small polar metabolites (identified based on their concentration/peak areas) include succinic acid (major component) followed by lactic acid > myo-inositol > scyllo-inositol > tyrosine > malic acid > talose > glucose > glycerol > phenylalanine. Galactose-6-phosphate was the least abundant metabolite present in the polar ESP fraction. 

Table 1 shows the retention times, peak areas, assigned individual masses, chemotaxonomy/chemical groups and Kyoto Encyclopedia of Genes and Genomes (KEGG) identities (ID), as well as the reported biological properties of each SM. The KEGG ID offers information on SM metabolism and biosynthesis pathways. These SM metabolites belong to 11 different classes including amino acids, amino sugars, amino acid lactams, organic acids, sugars, sugar alcohols, sugar phosphates, glycerophosphates, phosphate esters, disaccharides, and fatty acid derivatives. The highest number of compounds identified belong to the chemical class of organic acids (nine molecules), followed by amino acids (6 molecules), sugar alcohol (seven molecules), sugar (four molecules), sugar phosphate (two molecules), and other minor chemical groups. Interestingly, the major SM present (as putatively determined from the chromatographable peak areas) in the *D. caninum* ESP was succinic acid. 

#### Statistical Analysis Using MetaboAnalyst 4.0 Software 

We performed a statistical data analysis (univariate and multivariate) on the polar metabolites identified in Table 1. First, a univariate volcano plot analysis was performed to compare the datasets of the two groups (*D. caninum* versus culture media) and understand which metabolites are significantly different between the samples. The volcano plot also calculated fold change and *p*-values that enabled comparison of absolute value changes between the two-group mean (result plotted in log 2 scale to obtain same distance to zero baseline). The variable is reported here as significant if the value is above a given count threshold (>75% of pairs/variable), and has a fold change of >2 and a *p*-value of <0.05. Table 1 shows the fold change and the *p* values of these significantly different metabolites. For example, *D. caninum* ESP contains significant levels of succinic acid, meso-Erythritol, glycerol, and scyllo-Inositol 

The volcano plot showed that 18 metabolites of the *D. caninum* ESP were significantly different in their concentration from culture media-derived metabolites (fold change > 2, *p* value < 0.05) in their peak intensities (Figure 1).

In addition to univariate analysis, we performed multivariate chemometric analysis (applied to three sample groups) on the targeted matrix extracted, which included the identified metabolites. The PCA plot (Figure 2a) showed clear separation between the three sample groups of *D. caninum* ESP, glutamax medium control and pooled biological replicates of *D. caninum* ESP and culture media. Such clear separation of the groups indicated that helminth metabolites and culture media (glutamax) components could be readily distinguished and that *D. caninum* produces an array of metabolites when cultured in minimal medium. The partial least square discriminant analysis (PLS-DA) scores (Figure 2b) shows top 15 metabolites present in *D. caninum* ESP and medium. 

Hierarchical cluster analysis of the three sample groups produced a heat map of metabolites (Figure 3) that revealed the types of metabolites released into the media. The heat map shows distinct separation between the SM complements of the ESP compared to culture media. For example, while succinic acid, glycerol, and meso-erythritol were abundantly present in the ESP of *D. caninum,* these metabolites were not presented in the culture medium. On the other hand, fructose, ribitol, and maltose were presented in higher concentrations/intensities in culture medium, but were absent in *D. caninum* ESP. 

### 2.2. Non-Polar Metabolites of D. Caninum Identified Using GC-MS and LC-MS 

From the non-polar chloroform fractions of *D. caninum*, we identified 14 fatty acids ranging in lipidic carbon numbers C2–C22, including saturated, un-saturated and short chain fatty acids (SCFA) (Table 2). ^13^C18:0 was used as an internal standard and the reference standards (C10–C30) were used for validation. Fatty acids were identified by library matching techniques as described in the polar metabolites section above. Stearic acid (C18:0) is the major fatty acid present in the ESP. Tridecylic acid (C13:0) is the second major component of the non-polar fraction of D. caninum ESP. Three polyunsaturated fatty acids (oleic acid, linoleic acid, and arachidonic acid) and only one SCFA (acetate (C2:0)) were detected in the D. caninum SM ESP.

### 2.3. Literature Review Reveals Bioactive Small Molecules Present in the D. caninum ESP

Based on the literature and databases searches, we found that of 49 total metabolites identified (includes polar and lipid compounds), 12 of them (see Table 2) including glycerol [18], lactic acid [14], malic acid [15], methionine [16], octadecanoic acid [22], hexadacanoic acid [21], docosahexaenoic acid [23], acetate [24], and dodecanoic acid [20] were reported to have anti-inflammatory properties. Another four SM including citric acid [15], malic acid [15], aminobutyric acid [17], and fructose [19] were known to possess cardioprotective, anti-platelet aggregation, wound healing and pro-inflammatory activities, respectively. Dodecanoic acid [20] was also able to inhibit the growth of *Staphylococcus aureus*, *Bacillus cereus*, *Salmonella typhimurium,* and *Escherichia coli*.

## 3. Discussion

Gastrointestinal helminths affect humans, livestock and companion animals causing immense morbidity, growth retardation, and productivity losses, and yet very little is understood about the ESP of these helminths. ESP contains complex mixtures of compounds including macromolecules (inter alia proteins) and micromolecules (non-protein small molecules). Micromolecules or SM of the gastrointestinal helminths are largely unstudied and there is a paucity of information on metabolome composition of helminth ESP in general. 

Recent advances in metabolomics technologies, which can identify and quantify cellular metabolites using sophisticated analytical tools including MS and nuclear magnetic resonance (NMR) spectroscopy, have ushered in a new era of data mining and interpretation. However, metabolomics applications to study helminths and their interactions with vertebrate hosts is still in its infancy [9], with most of the previous studies devoted to understanding the effect of parasitic infections on host metabolite expression [25]. Only limited studies have been reported [9,26] on the metabolomes of blood flukes (*Schistosoma mansoni*) and gastrointestinal nematodes (*Ascaris lumbricoides* and *Ancylostoma caninum*), and until now there has been a paucity of information for tapeworms (cestodes). Other than fatty acids, none of the polar metabolites that were previously reported from *S. mansoni* and *A. lumbricoides* were detected here in *D. caninum* ESP. This variation in reported metabolite composition could be due to distinctiveness in the groups of helminths studied (spanning three different phyla), or differences in culture media, methods of ESP production, and metabolomics techniques used. It is known that in the free-living nematode *Caenorhabditis elegans*, the metabolism or expression of biochemicals is strongly dependent on diet and developmental stage [27]. 

Using a single component culture medium (glutamax), we have successfully cultured adult stage hookworms [13], whipworms, roundworms and now tapeworms ex vivo and collected their ESP. Recently, we reported 46 polar metabolites, 22 fatty acids and 5 SCFA from the somatic tissue extract of the dog hookworm *A. caninum*, and another 29 polar metabolites, 13 fatty acids and 6 SCFA from its ESP [13]. This current study on the ESP of *D. caninum* detected 35 polar metabolites with molecular weights (*m*/*z*) ranging from 89 to 342 atomic mass unit (amu), and 14 fatty acids with *m*/*z* ranging from 60–328 amu. While we detected six SCFA from hookworm ESP [13], only acetate was detected in the ESP of *D. caninum*. Unlike nematodes (*A. caninum*), cestodes such as *D. caninum* do not have an advanced internal digestive system and instead adsorb nutrients across the tegument. Interestingly, while pyroglutamic acid was the major compound detected in *A. caninum* ESP [13], succinic acid was the major compound detected in *D. caninum* ESP. Variations in major metabolic pathways can be expected since the worms belong to completely unrelated phyla. 

While the KEGG pathway analysis suggested that succinic acid is produced via tyrosine metabolism involving citrate cycle (TCA) pathways, pyroglutamic acid is produced via glutathione metabolism. Pyroglutamic acid is found in substantial amounts in brain, skin and plants [28]. Only one SCFA, acetate, was detected in the *D. caninum* ESP. The nature and origin of SCFA production/biosynthesis by helminths in general remains sketchy [13]. While it is possible that helminths synthesize SCFA de novo, the host microbiome could be a potential source of SCFA for tapeworms (assuming that antibiotic treatment did not completely kill all adhered bacteria upon removal from the canine host gut), since acetate, butyrate and propionate are known to be produced and utilized by bacteria [29]. Further studies will be needed to define the contribution of the commensal microbiome to SCFA synthesis and its detection in helminth ESP metabolites. 

Helminths clearly have negative health impacts on many infected vertebrate hosts, but the recent spotlight on iatrogenic helminth therapy [30,31] and the discovery of immunoregulatory proteins [32] and more recently metabolites [10,13] secreted by helminths has resulted in an appreciation of their pharmacopoeic properties. The ESP metabolites of *D. caninum* contain at least 12 SM that possess known bioactivities with relevance to human health. Of particular interest, the unsaturated fatty acid, docosahexaenoic acid has demonstrated anti-inflammatory activities [23], the SCFA, acetate is important in regulating colonic blood flow and ileal motility [33] and other metabolic processes that govern inflammatory processes. Novel targeted approaches for delivering the three major SCFAs—acetate, propionate and butyrate—to the gut is of great interest for the treatment of inflammatory bowel diseases [34] and given the acceptance of iatrogenic helminth infection, it is tempting to speculate that human infection with *D. caninum* presents an opportunity for targeted gastrointestinal delivery of therapeutic SCFAs. We believe that these anti-inflammatory SM, individually or in combination, are an integral part of the multi-pronged immunoregulatory storm released by parasitic helminths, and hold promise as a novel platform of therapeutics inspired by host-parasite coevolution.

## 4. Materials and Methods 

*D. caninum* (15–20 adult worms) was collected from stray dogs using methods we have described earlier for hookworms [13]. The worms were washed (5 times) with PBS containing 5% antibiotic/antimycotic (AA), transferred to pre-warmed culture media (2% glutamax in phosphate buffered saline, 2% AA) and were incubated for 6–7 h at 37 °C in 5% CO_2_. The ESP in culture media were collected, centrifuged at 1831× *g* for 20 min at 4 °C to remove cells/feces/debris, and the supernatant was filtered using 3 kDa cut off Amicon ultra centrifugal filters (Merck Millipore, Ireland, 15 mL) to obtain small molecule extracts. The samples were prepared for respective GC-MS and LC-MS analyses and were analyzed using the method described by us elsewhere [13]. 

### 4.1. GC-MS Analysis of ESP Samples of D. caninum

#### 4.1.1. Sample Preparation and Cryomill Extraction Methods

The *D. caninum* ESP (20 mg) was placed in chilled cryomill tubes (5 biological replicates for each worm), suspended in 600 µL of extraction solution of methanol:water (3:1, *v*/*v*) containing internal standard ^13^C-sorbitol (Sigma-Aldrich, Castle Hill, Australia), and then extracted using a Precellys 24 Cryolys unit (Bertin Technologies, avenue Ampère, France) at 6800 rpm (3 × 30 s pulses, 45 s interval between pulses) and −10 °C temperature (pre-chilled with liquid nitrogen). The homogenate (600 µL) was transferred to a clean microfuge tube (on ice) and 150 µL pre-chilled chloroform was added to it. A monophasic mixture of chloroform:methanol:water (1:3:1, *v*/*v*) was obtained by vortexing vigorously. The solution was chilled on ice for 10 min with regular mixing and then centrifuged for 5 min at 0 °C. The supernatant was transferred to a fresh 1.5 mL microfuge tube on ice and milli-Q water (300 µL) was added to each tube to obtain a biphasic partition of the solution (chloroform:methanol:water 1:3:3, *v*/*v*). The sample was vortexed vigorously and then centrifuged at 0 °C for a further 5 min. Both upper aqueous phase (methanol:water, 900 µL) and lower phase (chloroform) were derivatized and analyzed for polar and non-polar content respectively. A small amount of these samples were aliquoted for the pooled biological replicates (PBQC). The samples were grouped as ‘*D. caninum* ESP’, ‘PBQC quality control’ and ‘media control’.

#### 4.1.2. Derivatization and Analysis of Polar Fraction Using Targeted GC-MS Technique 

From the upper aqueous phase, we aliquoted 50 µL and transferred to a pulled point insert tubes and dried in an evaporator (Christ RVC 2-33 CD, John Morris Scientific, Chatswood, Australia) at 30 °C. Another 50 µL of the aqueous sample was added to the tubes every 30–40 min until completely dry. Anhydrous methanol (50 µL) was added to dehydrate the samples and then derivatized by adding 20 µL methoxyamine (30 mg/mL in pyridine, Sigma Aldrich, Castle Hill, Australia) at 37 °C for 30 min, followed by addition of 20 µL *N*,*O*-Bis(trimethylsilyl) trifluoroacetamide (BSTFA) + 1% trimethyl-chlorosilane (TMCS) (ThermoFisher Scientific Australia) at 37 °C for 30 min. It was then left standing at room temperature for 2 h and 1 µL of this derivatized polar sample was analyzed using an Agilent 7890 GC-MS (5973 MSD) [35]. Agilent VF-5 ms column (30 m × 0.25 mm × 0.25 µm) was used for chromatographic separation of metabolites. The oven was set to 35 °C, held for 2 min, then ramped at 25 °C/min to 325 °C and then held for 5 min. Helium was used as the carrier gas at a flow rate of 1 mL/min and the GC-MS metabolite *m*/*z* scanning range was set to 50–600 atomic mass unit (amu). Agilent’s Mass Hunter Quantitative Analysis software (v.7) was used for analyzing targeted metabolomics data and the Metabolomics Standard Initiative (MSI) level 1 was used for confirmation. Each compound was identified by a library matching technique, in which an observed EI-MS spectra [M^+^] of each metabolite was matched with EI-MS spectra in the Mass Hunter Library (MHL) as well as in the NIST library. The in-house Metabolomics Australia metabolite library (MAML) was used to extract target ion peak areas (in a *.csv* data matrix) for polar metabolites. The MAML was developed by running authentic standards in the GC-MS prior to running test samples. This same method was used for running our samples in the GC-MS. ^13^C-sorbitol and ^13^C^15^N-valine as were used as internal standards, which helped in determining the relative concentrations of each metabolite by comparing the metabolite peak areas with the peak areas of the internal standards.

#### 4.1.3. Analysis of Non-Polar Fraction Using Targeted GC-MS Approach

The non-polar fraction/organic phase of each biological replicate sample was sealed into GC vials and trans-esterified to create fatty acid methyl esters. Gerstel MPS2 autosampler robot was used for mixing the samples with 5 μL Methprep-II (Alltech). The solution was incubated for 30 min at 37 °C with slow shaking. Agilent 7890 gas chromatograph coupled to a triple quadrupole mass selective detector (Agilent Technologies, Australia) was used for sample analyses. Samples (2 L) were injected in split mode (10:1) into an inlet set at 250 °C. The chromatographic separation was achieved on a SGE BPX70 column (60 m × 0.25 mm i.d. × 0.25 um film thickness) (Trajan Scientific and Medical, Ringwood, Australia). The oven conditions were initially set to 70 °C for 1 min, then gradually raised to 150 °C (40 °C/min), 200 °C (4 °C/min), 220 °C (3 °C/min), 250 °C (°C/min) and finally held at 250 °C for 4 min. Helium was used as carrier gas (flow was constant at 1.5 mL/min) and the MS transfer line was set at 280 °C. Compounds were ionized using electron impact fragmentation (−70 eV) and mass spectra were collected in scan mode over the *m*/*z* range of 50–450 at 3.6 scans/s. Each compound ion was recorded in a positive mode (M^+^) and was identified by a library matching technique. The target ion peak areas were extracted using the in-house MHFAL and output as a data matrix in the required format (.csv) for further analysis. ^13^C18:0 was used as an internal standard. 

### 4.2. LC-MS Identification of SCFA

We used the LC-MS protocols described previously [13,36]. The ESP samples were extracted using the ‘Cryomill extraction method’ and a small portion (20 µL) of the extract was derivatized using 3-nitrophenylhydrazine (3-NPH), which converted SCFAs to their 3-nitrophenylhydrazones. The samples (1 µL) were then injected into a Shimadzu LC 30AD-TQ 8060 triple quadrupole mass spectrometer and analysed in triplicates. Authentic SCFA standards (including acetate, propionate, isobutyrate, butyrate, 2-methylbutyrate, isovalerate, valerate, 3-methylvalerate, isocaproate, and caproate) were run alongside samples in multiple reaction monitoring mode in order to facilitate Metabolomics Standard Initiative (MSI) level 1 confirmation. An Agilent EC-C18 poroshell 120 (50 mm × 2.1 mm × 2.7 µm column) was used for the analysis, using 100% water (+0.1% formic acid) mobile phase A and 100% acetonitrile (+0.1% formic acid) mobile phase B. The column flow rate was 0.55 mL/min, with the column temperature kept at 40 °C. The elution gradient was optimized at 15% B for 1 min, 15–25% B in 7 min, 25–30% in 1 min, 30–100% in 3 min, held at 100% for 3 min, then back to starting conditions and held for 2 min–providing a total run time of 17 min. The MRM transitions were optimized by direct infusion of the derivatives from a mixed standard solution containing each fatty acid. Collision energies were optimized (22 eV, 25 eV, and 28 eV) for each analyte and the most sensitive energies were selected for the analysis. Each ion was recorded in positive mode [M^+^] and were identified as described above. 

### 4.3. Data Analyses and Statistical Interpretation

The GC-MS data was analyzed in a targeted approach using Agilent’s Mass Hunter Quantitative Analysis software (v.7). Target ion peak areas for polar and non-polar metabolites were extracted using the in-house Metabolomics Australia metabolite library and output as a data matrix (in .csv format) for further analysis. Chemometric and statistical analyses were undertaken using MetaboAnalyst 4 [37]. The GC-MS data was normalised with respect to median and log transformed prior to performing chemometric analyses including Principal Component Analyses (PCA), Partial Least Squares-Discriminant Analysis (PLS-DA) and cluster heatmap analyses (compound concentration versus samples in 5 replicates).

### 4.4. Literature Searches and Their Content Analyses Focusing on Anti-Inflammatory Properties

The small molecules identified from the *D. caninum* ESP were queried using: (1) Human metabolome database (HMDB), which contains 114,100 metabolite entries including both water-soluble and lipid soluble metabolites [28]; (2) DrugBank, which contains information on 2280 drug metabolites [38]; and (3) PubChem, which contains live record counts of 94,017,529 compounds (mostly small molecules) with data on chemical structures, identifiers, chemical, and physical properties, biological activities, patents, health, safety, toxicity data, and other properties [39]. In addition, search engines including Google Scholar and SciFinder Scholar were used for tracing the references on biological activities of each compound. Unique search terms/keywords including “biological activities”, “anti-inflammatory”, and “immune regulation” were used. Content analyses of these databases and references were performed focusing on the reported biological activities, and this information was then tabulated and cited against each compound. 

## 5. Conclusions

We have shown that *D. caninum* excrete/secrete as many as 35 known polar metabolites and 14 fatty acids, and that GC-MS and LC-MS can be used to profile the metabolite complement of tapeworm ESP. While most of the polar metabolites belong to the chemotaxonomy of organic acids (with nine small molecules), most of the non-polar metabolites were saturated fatty acids. Four unsaturated fatty acids and one SCFA were also detected in the non-polar faction of *D. caninum* ESP. PCA analysis showed clear separation of *D. caninum* secreted metabolites from culture media components, and the major metabolite was identified as succinic acid. Of a total of 49 metabolites identified, 12 have known pharmacological properties, notably as anti-inflammatory agents. These bioactive molecules may be, individually or in combination, responsible for the immune dampening effects of tapeworms in their definitive mammalian hosts, allowing them to survive in such hostile environments as the gut. Future work will entail purification and isolation of pharmacologically bioactive compounds from *D. caninum* ESP and assessment of their anti-inflammatory properties at physiologically relevant concentrations in appropriate in vitro and in vivo settings.

## Figures and Tables

**Figure 1 molecules-24-01480-f001:**
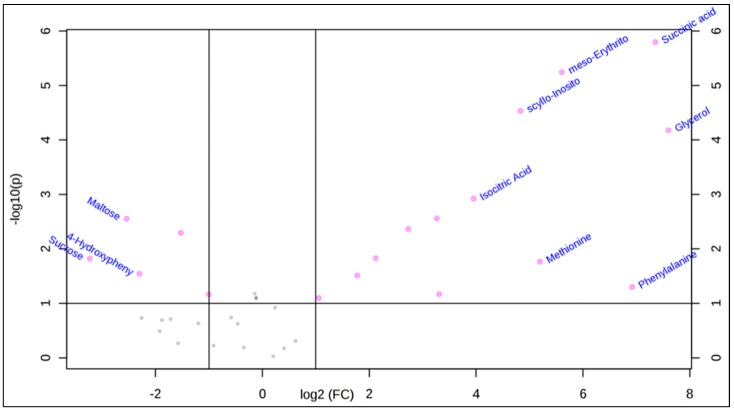
Univariate volcano plot analysis with fold change threshold (x) 2 and t-tests threshold (y) 0.1 showing metabolites with significantly different abundance between *D. caninum* ESP and culture medium. The further the metabolite position is from (0,0), the more significant the difference in metabolite presence (pink dots). Data analyzed by MetaboAnalyst 4 (samples normalized by mean, log transformation and autoscaling).

**Figure 2 molecules-24-01480-f002:**
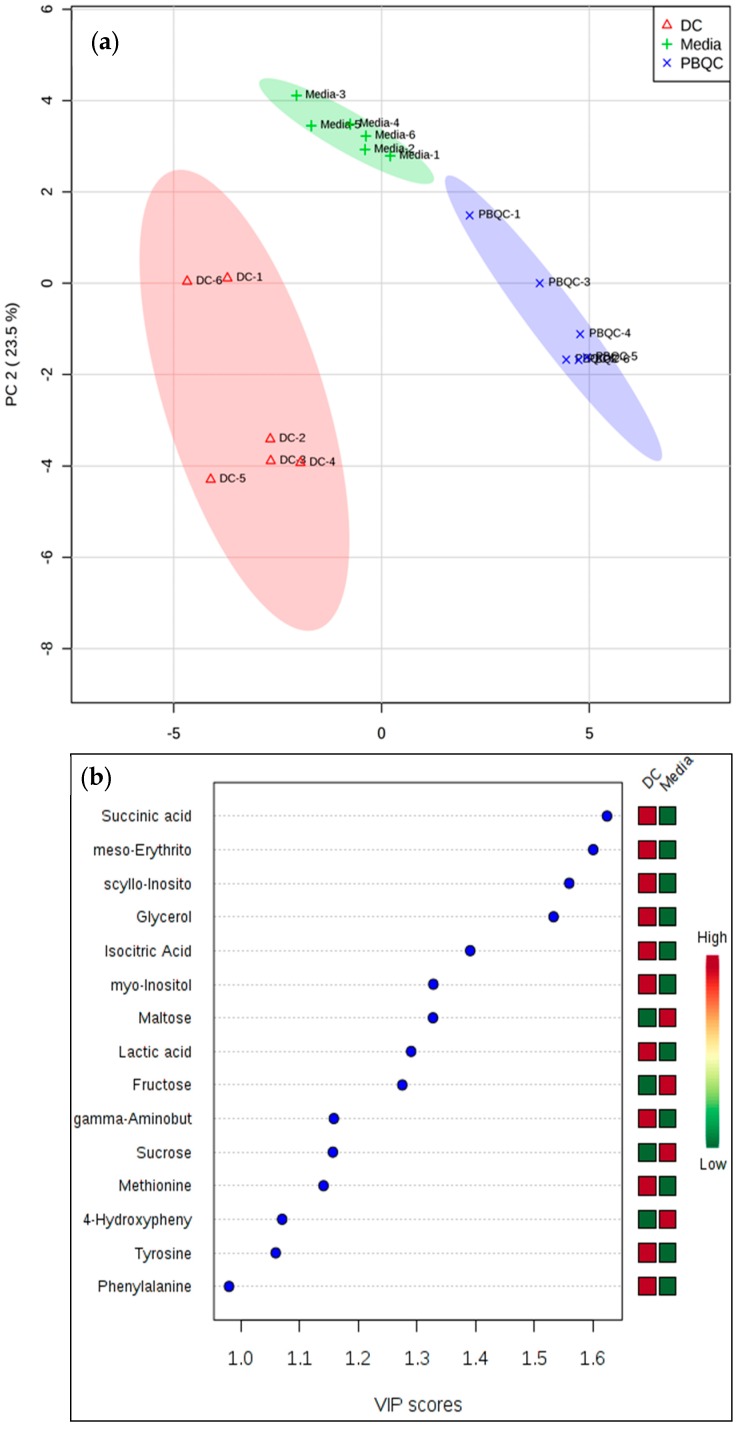
Multivariate principal component analysis (PCA) of *D. caninum* ESP. (**a**) PCA scores plot showing clear separation of three sample groups (*D. caninum*, media control and pooled biological replicates of ESP and medium samples). (**b**) Important features identified by PLS-DA. The colored boxes on the right indicate the relative peak intensities of the corresponding metabolite in *D. caninum* ESP and culture medium (15 important SM). Data analyzed by MetaboAnalyst 4 (samples normalized by mean, log transformation, and autoscaling).

**Figure 3 molecules-24-01480-f003:**
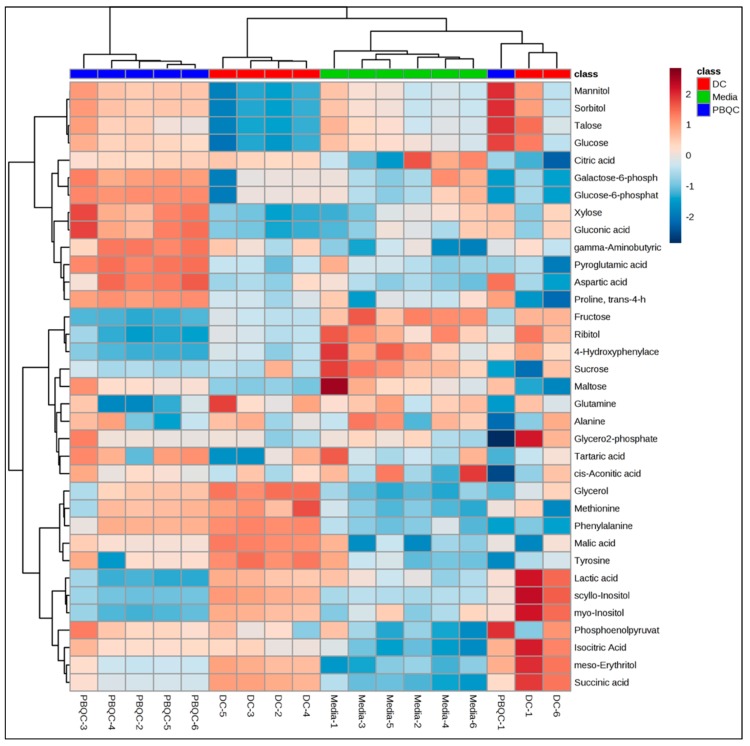
Heat map showing polar metabolites of *D. caninum* ESP. Heat map was generated using MetaboAnalyst 4 (*N* = 6 replicates). Deep brown denotes highest peak areas of metabolites and deep blue denotes lowest peak areas or complete absence of metabolites.

**Table 1 molecules-24-01480-t001:** Total polar small molecule (SM) identified by gas chromatography mass spectrometry (GC-MS) from the excretory-secretory products (ESP) of *D. caninum* with their molecular characteristics.

Compounds	Rt *	Peak Area	Log 2 (FC) **	Mass (*m*/*z,* M^+^)	Chemical Class ***	KEGG ID ****	Reported Bioactivities
Lactic acid	6.73	4,919,948	2.73	90	Organic acid	C00186	Anti-inflammatory [14]
Alanine	6.92	39,820	ns	89	Amino acid	C00041	N/A
Succinic acid	8.11	14,637,634	7.35	118	Organic acid	C00042	N/A
Malic acid	8.97	80,527	3.30	134	Organic acid	C03668	Anti-platelet & anti-inflammatory [15]
Meso-Erythritol	9.02	24,013	5.60	122	Sugar alcohol	C00503	N/A
Aspartic acid	9.20	217	ns	133	Amino acid	C00402	N/A
Methionine	9.22	7437	5.19	149	Amino acid	C00073	Anti-inflammatory [16]
trans-4-Hydroxyproline	9.23	1874	ns	131	Amino acid	C01157	N/A
Pyroglutamic acid	9.27	28,950	ns	129	Amino acid lactam	C01879	N/A
GABA	9.28	2666	2.12	103	Fatty acid derivative	D00058	Wound healing & neurotransmitter [17]
Phenylalanine	9.43	45,266	6.92	165	Amino acid	C00079	N/A
Phosphoenol-pyruvate	9.53	261	1.05	168	Phosphate ester	C00074	N/A
Tartaric acid	9.73	311	ns	150	Organic acid	C02107	N/A
Xylose	9.86	5285	ns	150	Sugar	C00181	N/A
Ribitol	10.08	5464	−1.00	152	Sugar alcohol	C00474	N/A
Glycerol-2-phosphate	10.11	6003	ns	172	Glycerophosphate	C02979	N/A
Glycerol	10.13	47,846	7.59	92	Sugar alcohol	C00116	Anti-inflammatory [18]
cis-Aconitic acid	10.27	173	ns	174	Organic acid	C00417	N/A
Citric acid	10.53	757	ns	192	Organic acid	C00158	Cardioprotective [15]
Isocitric acid	10.54	872	3.95	192	Organic acid	C00311	N/A
Fructose	10.73	17,559	−1.52	180	Sugar	C02336	Pro-inflammatory [19]
Talose	10.80	70,147	ns	180	Sugar	N/A	N/A
Glucose	10.83	66,453	ns	180	Sugar	C00031	N/A
Mannitol	10.94	9570	ns	182	Sugar alcohol	C00392	N/A
Tyrosine	10.96	89,216	1.77	181	Amino acid	C00082	N/A
Sorbitol	10.98	9527	ns	182	Sugar alcohol	C00794	N/A
Glutamine	11.21	657	ns	146	Organic acid	C00064	
Gluconic acid	11.27	5178	ns	196	Organic acid	C00257	N/A
Scyllo-Inositol	11.38	92,797	4.83	180	Sugar alcohol	C06153	N/A
4-Hydroxy-Phenylacetic acid	11.42	678	−2.30	152	Benzenoid	C00642	N/A
Myo-Inositol	11.67	430,582	3.26	180	Sugar alcohol	C00137	N/A
Galactose-6-Phosphate	12.32	119	ns	260	Sugar phosphate	N/A	N/A
Glucose-6-Phosphate	12.41	193	ns	260	Sugar phosphate	C00092	N/A
Sucrose	13.40	8792	−3.23	342	Disaccharide	C00089	N/A
Maltose	13.61	5089	−2.54	342	Sugar	C00208	N/A

* Rt = retention time in minute. ** log 2 of volcano plot (Metaboanalyst 4) represent the mean ratio fold change plotted on a log 2 scale (log transformed) of the relative abundance of each metabolite between two samples, so that same fold change (up/down regulated or +/−) will have the same distance to the zero baseline. The values were taken from and shows the significant metabolites present in the ESP. *** chemical class was taken from HMDB (http://www.hmdb.ca/). **** KEGG ID (https://www.genome.jp/kegg/) provides the information on the biosynthetic and metabolic pathways of a compound. ID = identity; ns = not significant; FC = fold change; N/A = not available.

**Table 2 molecules-24-01480-t002:** Fatty acids identified from the ESP of *D. caninum* with their lipid number, mass, KEGG identities, and reported biological properties.

Fatty Acid Names. (Lipid Number)	Rt *	Peak Areas	Mass (*m*/*z,* M^+^)	KEGG ID **	Reported Biological Properties
Undecanoic acid (C11:0)	7.35	5503	186	C17715	N/A
Decanoic acid (C10:0)	7.67	5507	172	C01571	N/A
Tridecylic acid (C13:0)	9.03	21,853	214	N/A	N/A
Dodecanoic acid (C12:0)	9.22	767	200	C02679	Anti-inflammatory & antibacterial [20]
Tetradecanoic acid (C14:0)	11.29	1864	228	C06424	N/A
Pentadecylic acid (C15:0)	12.53	3431	242	C16537	N/A
Hexadecanoic acid (C16:0)	13.84	16,171	256	C00249	Anti-inflammatory [21]
Heptadecanoic acid (C17:0)	15.25	1238	270	N/A	N/A
Octadecanoic acid (C18:0)	15.25	32,883	284	C01530	Anti-inflammatory [22]
Oleic acid (C18:1, 9Z- cis)	17.16	1219	282	C00712	N/A
Arachidonic acid (C20:4)	18.15	2076	304	C00219	N/A
Linoleic acid (C18:2)	18.33	5336	280	C01595	N/A
Eicosanoic acid (C20:0)	19.60	5981	312	C06425	N/A
Docosahexaenoic acid (C22:6)	26.84	6960	328	C06429	Anti-inflammatory [23]

* Rt = retention time in minute. ** KEGG ID (https://www.genome.jp/kegg/) provides the information on the biosynthetic and metabolic pathways of a compound. ID = identity; ns = not significant; FC = fold change; N/A = not available.

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
