# Peer review of "Characterization of Tapeworm Metabolites and Their Reported Biological Activities"

_molecules, 2019, doi:10.3390/molecules24081480_

Round 1

Reviewer 1 Report

Excretory-secretory products from Dipylidium caninum were collected and analyzed. Results are clearly presented and documented. Conclusions are supported by experimental data and literature research.

Author Response

We thank the reviewer for their highly positive comments.

Reviewer 2 Report

The authors used the GC-MS and LC-MS methods to identify metabolites. Therefore, the authors must present results regarding the validation of the GC-MS and LC-MS methods.

Author Response

We thank the reviewer for his/her very positive comments. We have now given more details on the validation of GC-MS and LC-MS methods in our revised ms.

Reviewer 3 Report

The Manuscript reports on an identyfication of small molecules excreted by tapeworm species Dipylidium caninum. The overall quality of the manuscript is good; however several issues have to be addressed prior to its final acceptance for publication.

1.       Title of manuscript ‘Characterization of tapeworm metabolites using mass spectrometry reveals bioactive small molecules’ suggests that mass spectrometry was the main analytic technique used for the determination of those metabolites. However, text hardly says anything about MS analysis performed. In fact, there are only m/z values in table 1 and 2, but there is no specific information about spectrometers used, ion sources, ionization modes, type of mass analyzers, type of ions recorded ( [M+H]+, [M-H]-, adducts) or analysis conditions. However, authors spend a lot of time for databases search and statistical interpretation of obtained results thus I suggest to think about the change of the title to more adequate to the results presented.

2.       Material and methods section lacks conditions of some experiments, for example, mentioned above mass spectrometry analysis. Additionally, I see no paragraph regarding conditions of liquid chromatography experiments. In fact, there is no word about the type of liquid chromatography authors used. Was it HPLC or some other? What was the column used, what type of solvents were applied for elution? Did authors use any other detector aside from mass spectrometer? It’s hard to compare retention time of signals obtained by authors with data in databases when there is no specific information about the conditions in which those were recorded.

3.       m/z values and concentration/peak areas alone are not enough to a proper determination of isolated compounds, especially while working on complexed, biological samples. Did authors use any reference substances to confirm proposed results?

4.       Table 1 and 2 need to be unified. Tab 1 is sorted by compounds name while Tab 2 by retention time. Chromatography results are usually sorted by increasing retention time of isolated compounds, so I think both tables should be organized the same way. Moreover, retention time needs a unit, and please pay attention to significant figures in both tables. Additionally, type of ions recorded should be specified in the table header, for example m/z [M+H]+ or m/z [M-H]-.

Successful implementation of improvements listed above followed by resubmission of the manuscript should enable the possibility for its publication in Molecules journal.

Author Response

The Manuscript reports on an identification of small molecules excreted by tapeworm species Dipylidium caninum. The overall quality of the manuscript is good; however several issues have to be addressed prior to its final acceptance for publication.

1.       Title of manuscript ‘Characterization of tapeworm metabolites using mass spectrometry reveals bioactive small molecules’ suggests that mass spectrometry was the main analytic technique used for the determination of those metabolites. However, text hardly says anything about MS analysis performed. In fact, there are only m/z values in table 1 and 2, but there is no specific information about spectrometers used, ion sources, ionization modes, type of mass analyzers, type of ions recorded ([M+H]+, [M-H]-, adducts) or analysis conditions. However, authors spend a lot of time for databases search and statistical interpretation of obtained results thus I suggest to think about the change of the title to more adequate to the results presented.

Ans: We thank the reviewer for their constructive comments and suggestions. We have now incorporated all the information suggested by the reviewer in our revised manuscript (method section) and also changed the title of the ms.

2.       Material and methods section lacks conditions of some experiments, for example, mentioned above mass spectrometry analysis. Additionally, I see no paragraph regarding conditions of liquid chromatography experiments. In fact, there is no word about the type of liquid chromatography authors used. Was it HPLC or some other? What was the column used, what type of solvents were applied for elution? Did authors use any other detector aside from mass spectrometer? It’s hard to compare retention time of signals obtained by authors with data in databases when there is no specific information about the conditions in which those were recorded.

Ans: We thank the reviewer for spotting this critical error and making great suggestions. We are sorry for missing out the information on the LCMS method. It may have fallen of the text during the corrections/edition process. We have now given full description and condition off the LCMS method in our revised ms.

3.       m/z values and concentration/peak areas alone are not enough to a proper determination of isolated compounds, especially while working on complexed, biological samples. Did authors use any reference substances to confirm proposed results?

Ans: We thank the reviewer for their suggestions. Yes, we used reference standards to confirm the proposed results. We have now given full details on how the identification were performed using inhouse compound library, NIST database and reference standards.

4.       Table 1 and 2 need to be unified. Tab 1 is sorted by compounds name while Tab 2 by retention time. Chromatography results are usually sorted by increasing retention time of isolated compounds, so I think both tables should be organized the same way. Moreover, retention time needs a unit, and please pay attention to significant figures in both tables. Additionally, type of ions recorded should be specified in the table header, for example m/z [M+H]+ or m/z [M-H]-.

Ans: We thank the reviewer for their suggestions. We have made changes to the tables as recommended.

Successful implementation of improvements listed above followed by resubmission of the manuscript should enable the possibility for its publication in Molecules journal.

Ans: We thank the reviewer for this recommendation.

Round 2

Reviewer 3 Report

Thank you for your responses and all changes made in the manuscript.

One little thing. Move description of 'Table 2' above the table (line 191).